# The Impact of Urban Facilities on Crime during the Pre- and Pandemic Periods: A Practical Study in Beijing

**DOI:** 10.3390/ijerph20032163

**Published:** 2023-01-25

**Authors:** Xinyu Zhang, Peng Chen

**Affiliations:** School for Informatics Engineering and Cyber Security, People’s Public Security University of China, Beijing 100038, China

**Keywords:** COVID-19, residential burglary, non-motor vehicle theft, urban facility, negative binomial regression, geographical weight regression, Beijing

## Abstract

The measures in the fight against COVID-19 have reshaped the functions of urban facilities, which might cause the associated crimes to vary with the occurrence of the pandemic. This paper aimed to study this phenomenon by conducting quantitative research. By treating the area under the jurisdiction of the police station (AJPS) as spatial units, the residential burglary and non-motor vehicle theft that occurred during the first-level response to the public health emergencies (pandemic) period in 2020 and the corresponding temporal window (pre-pandemic) in 2019 were collected and a practical study to Beijing was made. The impact of urban facilities on crimes during both periods was analyzed independently by using negative binomial regression (NBR) and geographical weight regression (GWR). The findings demonstrated that during the pandemic period, a reduction in the count and spatial concentration of both property crimes were observed, and the impact of facilities on crime changed. Some facilities lost their impact on crime during the pandemic period, while other facilities played a significant role in generating crime. Additionally, the variables that always kept a stable significant impact on crime during the pre- and pandemic periods demonstrated a heterogeneous impact in space and experienced some variations across the periods. The study proved that the strategies in the fight against COVID-19 changed the impact of urban facilities on crime occurrence, which deeply reshaped the crime patterns.

## 1. Introduction

The sudden outbreak of COVID-19 caused a huge impact on social life. In order to curb the spread of the virus, countries and regions all over the world adopted a series of prevention and control measures including lockdowns, social distancing, stay-at-home orders, etc. However, the implemented measures simultaneously generated a versatile impact on offending activities in the city. Recently, many studies have contributed to uncovering the impact of COVID-19 prevention and control on crime. The common conclusions conducted from these studies are that the pandemic-related prevention and control measures induced a reduction in general crime [1,2]. Lately, as more work has focused on this phenomenon, people have found that the crime variation in smaller spatial units is heterogeneous [3,4,5,6]. In order to explain the variation in space, scholars have included social-economic variables such as population mobility changes [7,8], population racial characteristics [9,10,11], and economic variables [12,13]. However, some limitations in the current research still exist as to whether the impact of urban facilities, especially typical crime attractors [14], on crime varied by the pandemic is little understood. Being the first country to report cases of COVID-19, China initiated a series of prevention and control measures including home isolation and social distancing at the early stage. Consequently, most people stayed at home for working or studying, the business in shopping and service centers stopped, schools were closed, etc., which also changed the opportunity conditions of crime in urban settings. Thus, investigating how urban facilities impacted crime occurrence during the pre- and pandemic periods should be further investigated.

This paper plans to conduct an empirical study on residential burglary and non-motor vehicle theft that occurred during the pre- and pandemic periods in Beijing and tries to uncover whether the associations between occurred crimes and urban facilities across the periods changed. The rest of this paper is organized as follows. First, the relevant research associated with this study is summarized, then the data and methods are described. After that, the corresponding analysis results are shown, and finally, the major conclusions and prospects of future work are presented.

## 2. Related Studies

The COVID-19 pandemic provided an opportunity for a rare natural experiment, allowing scholars from various countries and regions to examine the potential impact of pandemic prevention and controls on crime in urban settings. The initial research focused on describing the crime variation by time in specific cities and found that most crimes in different countries and cities had been observed to reduce. For example, many studies proved that residential burglary was significantly reduced during the pandemic period [1,15,16,17,18,19,20], but in some areas, the reduction in residential burglary was not significant [21]. Following that, the effect of the pandemic on other crimes was studied and some complicated and interesting results were observed [22]. For example, it was discovered that some crimes other than residential burglary increased during the pandemic period [23,24]. Additionally, while vehicle theft was frequently observed to decrease in some cities during the pandemic period [15,16,19,24,25,26,27], an increase also existed in some cities [1,23], for example, in Los Angeles, motor vehicle theft even showed no significant variation during the pandemic days [21]. Aside from theft-related crimes, other types of crime have also shown complex variation patterns by cities. For example, Mcdonald and Balkin [18] found that rape in San Francisco and Philadelphia decreased during the pandemic period but increased in Los Angeles and New York. A reduction in robbery was observed in many cities [18,21,25,28,29], but the change was not significant in Sweden [17], which may be due to the less stringent local pandemic prevention and control measures. Moreover, a study of 28 large U.S. cities found no significant change in assaults in public places and serious assaults at homes [19], but studies in Sweden [17] and Japan [30] showed a downward trend.

After experiencing the study on crime variation by cities in the early stage of the pandemic, people have begun to continuously investigate crime variation in smaller spatial units. For example, Campedelli et al. [3] found that the reduction in crime in Chicago, USA varied by neighborhood and crime type following the implementation of COVID-19 containment policies, and it was thought that the differences between residential and non-residential distribution may be responsible for this pattern. Felson et al. [4] conducted a study of 879 blocks in Detroit, USA and found that burglaries shifted from primary residential areas to mixed residential–commercial areas during the pandemic. Additionally, another study in Chicago found that crime declined but that the crime clusters changed during the pandemic [6]. García-Tejeda et al. [5] found a decrease in shootings in Mexico during the pandemic, but a shift in spatial agglomeration was also observed. Evidence from the Indonesian city of Makassar showed that the lockdown measures led to an increase in burglary, but more burglaries occurred at home and on the streets and fewer occurred in commercial areas [31]. Dewinter et al. [32] analyzed the changes in crime in the streets of Antwerp, Belgium during the pandemic and found that although crime decreased, the variation in crime distribution on different streets showed heterogeneous patterns. Sun et al. [33] analyzed the correlation between the crime locations and the COVID-19 case locations in London and found that the higher the infection rate, the lower the robbery, burglary, theft, and handle rate. Payne and Langfield [34] studied drug-related crimes in Queensland, Australia, and found that drug crimes increased significantly during the pandemic period, and the drug trading market demonstrated significant spatial displacement. Compared with the research completed at the early stage of the pandemic, although the following studies focused on crime patterns in smaller spatial units and extended knowledge of crime variation by the impact of the pandemic, more descriptive analyses rather than quantitative analyses were performed, which suggests that the understanding of crime variation by the impact of the pandemic in space is not enough.

In recent studies, people have started to include social and economic variables to interpret the spatial distribution of crime variation in the context of COVID-19 prevention and control. The variables include population mobility, ethnic characteristics, and economic factors et al. For example, Estevez-Soto [7] conducted an analysis using public transport data and crime data in Mexico City and found that a positive relationship existed between mobility and crime rates during the pandemic. Meanwhile, a study conducted in the United Kingdom also proved this phenomenon [8]. Cheung and Gunby [35] analyzed the Google mobility data and crime in New Zealand during the pandemic period and found that the changes in mobility patterns were significantly correlated with crime rate variation, specifically, the increase in mobility in residential areas was significantly correlated with the decrease in property crime in both residential and non-residential areas. With regard to the racial characteristics of the population, Moise and Piquero [10] investigated domestic violence in Miami-Dade County during the lockdown period and found higher crime rates were associated with Black or African-Americans. Semukhina’s [11] study in Texas, USA also found that the crime committed by African-Americans did not decline during the pandemic period, but the crime rate of females was higher than that of males. Bullinger et al. [9] also conducted a neighborhood-level analysis of domestic violence in Chicago during stay-at-home order days and found that the areas with a higher proportion of tenants had more domestic violence. In terms of economic factors, the evidence from India [12] and the United States [13] suggest that the lockdowns le=d to higher unemployment, which in turn resulted in higher crime rates. However, in India, the concentrated crimes were robbery and other property offenses, while in the United States, the studied crimes were shootings and violent offences. A study from Canada that finished during the pandemic measured the crime changes in the Saxton region by differentiating economic levels and discovered a general decline in crime in the CBD areas but the opposite trend in poorer areas [36]. Ceccato et al. [37] conducted a comparative study of New York in the United States, Sao Paulo in Brazil, and Stockholm in Sweden, and proved that different restrictive policies would lead to crime varying by geographic location and economic development levels.

Thus far, people have conducted a great many descriptive and quantitative studies on crime in the context of COVID-19 prevention and control. At present, investigating crime’s variation in space from the social and economic perspectives has received more attention, as including the social and economic backcloth could help enhance the understanding of crime variation in space by the impact of the pandemic. However, some shortcomings still exist in the current research, as the social and economic variables have mainly focused on population, unemployment, etc., but less attention has been paid to urban facilities. As a matter of fact, urban facilities play important roles in attracting or generating crime opportunities [14], and some typical and specific facilities such as train or subway stations, bars, pubs, nightclubs, parks, hospitals, markets, etc. all have a potential impact on the occurrence of property or violent offenses [38,39,40,41]. A series of previous studies have proved that pandemic prevention and control measures will have a profound impact on people’s daily lives, which would significantly change the routine behavior patterns of people in using urban facilities. For example, compulsory staying at home for working and learning would reduce the density and flow of people in entertainment places, pubs, bars, schools, shops, etc., which will inevitably have a positive/negative impact on the generation of crime opportunities. Therefore, this paper will conduct research on crime distribution associated with urban facilities during the pre- and pandemic periods, and seek to find how crime varied by the type of facility.

## 3. Study Area and COVID-19 Prevention and Control Measures

The study area included the main urban districts of Beijing, which administratively includes Xicheng, Dongcheng, Chaoyang, Haidian, Fengtai, and Shijingshan Districts. According to the Annual Statistic Report in 2020 (https://nj.tjj.beijing.gov.cn/nj/main/2020-tjnj/zk/indexch.htm, accessed on 1 November 2022), the main urban districts are about 1385 km^2^, and the population living in the main urban districts is about 10.988 million. The Beijing government officially confirmed the discovery of the first COVID-19 case on 20 January 2020, and then launched a first-level response to the public health emergencies (FRPHE) to cope with the challenges of the COVID-19 virus spreading on 24 January. After maintaining a series of prevention and control measures for 14 weeks, Beijing officially announced that the spread of the COVID-19 virus had been controlled and from 30 April 2020 (http://www.beijing.gov.cn/fuwu/bmfw/wsfw/ggts/202004/t20200429_1888375.html, accessed on 1 November 2022), the emergency response was adjusted from the first-level to the second-level. Figure 1 shows the count of officially reported infection cases in Beijing during the FRPHE period (data from Beijing Municipal Health Commission: http://wjw.beijing.gov.cn/, accessed on 1 November 2022). It could be recognized that at the end of January, the cumulative number of confirmed cases rose rapidly (red line). The first “turning point” appeared in mid-February, when the number of daily reported confirmed cases began to decline (blue line). At the second “turning point”, the cumulative curve kept flattening (red line), and the daily reported confirmed cases kept declining (blue line). By 30 April, the cumulative number of cured cases (green line) approached the cumulative number of confirmed cases (red line), and the daily reported confirmed cases were reduced to zero level, which indicates that the spread of the COVID-19 virus in Beijing was successfully controlled.

During the FRPHE period in Beijing, a series of specific measures fighting against the spread of COVID-19 virus was implemented, where the typical strategies included stay-at-home orders and social distancing. Furthermore, people’s routine activities in urban facilities were also restricted. The facilities include public transportation, municipal industries, finances and enterprises, retails and supermarkets, hotels and motels, medical services, education agencies, residential areas, entertainment places, restaurants, green lands and parks, etc. The restrictions on the facilities are shown in Table 1.

## 4. Data and Methods

### 4.1. Data

#### 4.1.1. Crime Data

The crime data for this study came from the Beijing Municipal Public Security Bureau (BMPSB), and the crime types used for research were residential burglary and non-motor vehicle theft. According to the Penalty Code in China, residential burglary refers to the offense of breaking into the owner of the room without permission, and includes those that occurred in residential areas and merchants. Being a typical property crime, residential burglary has received much attention [42,43]. Non-motor vehicle theft refers to the offense of stealing electric bicycles (including batteries), bicycles, etc. outdoors. The information embedded in the crime dataset includes the registered number, occurring date and time, and occurring address. The crimes that occurred during the FRPHE period (between 24 January 2020 and 29 April 2020) and were located within the urban districts were collected for the study. In order to ensure that the crime pattern during the pandemic period could be comparable, the crime recordings that occurred during the corresponding period and areas in history were also collected (between 24 January and 29 April 2019). Table 2 presents the descriptive statistics of the crime data. From these, it can be seem that both residential burglary and non-motor vehicle theft were apparently reduced during the pandemic periods in 2020.

#### 4.1.2. Independent Variables

The urban facilities used for analysis were from the POI data, which was openly accessed from AMap (AMap is a comprehensive platform that provides digital map content, navigation, and location service solutions, the weblink of the AMap is www.amap.com) platform. A total of 29 types of POIs were collected and treated as potential attractors of property crime [44]. In order to conduct the spatial analysis, the area under the jurisdiction of the police station (AJPS) in Beijing was treated as the spatial unit. The reason for using the AJPS instead of the uniform grid is that the crime, police resources, population, facilities, etc. were registered and managed by the AJPS, hence, aggregating the data by the AJPS could remove the bias of aggregating by the uniform grid and make the analysis more acceptable. The Shpfile data of the AJPS was accessed from the BMPSB and there are 183 AJPSs in the main urban districts of Beijing available for analysis. The basic descriptive statistics of the POIs by AJPS units are shown in Table 3. Furthermore, the road network and subway station flow data were collected and treated as the control variables. The road network data was accessed from OpenStreetMap (OSM) through API and categorized into three different types according to its transportation function, namely, vehicles, bicycles and pedestrians, and pedestrians. The density of the three types of road network was computed by overlaying the data with AJPS units and treated as the local environment complexity. The subway station flow data originated from the Beijing Municipal Commission of Transportation (BMCT) and was aggregated by AJPS units to measure the local population mobility. As the subway stations were not distributed uniformly in the urban districts of Beijing, the influence of the subway flow on the neighboring AJPSs that have no subway stops could be estimated using the Kernel density method.

### 4.2. Methods

#### 4.2.1. Negative Binomial Regression (NBR)

The NBR model was used to analyze the relationship between the facilities and crime during the pre- and pandemic periods. NBR is a mixed Poisson regression model. For the interpretation of count variables, Poisson regression is used in the condition that the mean and variance of the dependent variables are equal. However, when the variance is greater than the mean value, the data are over-dispersed and NBR is required. The number of crimes conforms to the characteristics of a count model, so NBR is broadly used in analyzing changes of crime pattern in space [13,45,46]. The principle of NBR is:$$Pr(Y=y)=\frac{\Gamma (\tau +y)}{y!\Gamma (\tau )}{\left(\frac{\tau }{\tau +\mu }\right)}^{\tau }{\left(\frac{\mu }{\mu +\tau }\right)}^{y}\;y=0,1,\ldots ,n;\mu ,\tau >0$$
where *y* is the dependent variable; *Pr* is the probability of getting *y* in the sampling unit; Γ is the Gamma function; *μ* is the mean; *τ* is the ‘shape’ parameter; the variance of *Yy* is *μ* + *μ^2^*/τ.

#### 4.2.2. Geographic Weight Regression (GWR)

GWR is a local regression model that allows for spatial variation in the estimated parameters: one for each areal unit (AJPS). Once the appropriate global model was identified, GWR models were estimated for both types of crime. GWR can be represented using the following equation:$$y_{i}=\beta_{0}\left(u_{i},v_{i}\right)+\sum_{k}\beta_{k}\left(u_{i},v_{i}\right)x_{ik}+\varepsilon_{i}$$
where *y_i_* represents the value for a crime type at location *i*; *β_0_(u_i_, v_i_)* represents the constant for location *i*; *β_k_(u_i_, v_i_)* represents the estimated parameter for independent variable *x_k_* at location *i*; and *ε_i_* is the independently and identically distributed residual at location *i* [47,48,49].

## 5. Results and Discussion

### 5.1. Crime Variations in Space and Time

First, the basic temporal and spatial patterns of residential burglary and non-motor vehicle theft during the pre- and pandemic periods were investigated and compared. The boxplots of the crime count by day during the two periods in Beijing were generated. As shown in Figure 2, it can be seen that the residential burglary and non-motor vehicle theft that occurred in Beijing apparently decreased during the pandemic period. The independent sample t-test was run to examine the difference, and the results indicated that differences were all significant at the *p* < 0.001 level, which reflects that the pandemic posed a significant negative effect on the occurrence of property crime.

Figure 3 demonstrates the spatial distribution of both crimes in space during the pre- and pandemic periods. From the result, it could be observed that the crime patterns in space were seriously reshaped. First, according to the spatial statistic of Global Moran’s I in Table 4, it can be seen that the clustering effect of both residential burglary and non-motor vehicle theft were reduced from the pre- to pandemic period, which implies that the pandemic prevention measures not only depressed the crime count, but also lowered the spatial concentration. Second, most hot areas generated during the pre-pandemic period diminished or even disappeared in the pandemic period. In order to better understand the crime pattern in space, quantitative analysis was performed on both types of crime in the 183 AJPS units, and the statistic distribution of the increase/decrease is shown in Figure 4. It was found that during the pandemic period, the count of both crimes in most AJPS units decreased, while in some units, the crimes increased. Specifically, residential burglary decreased in 119 AJPSs, was unchanged in 53 AJPSs, and increased in 11 AJPSs (Figure 4). An extreme pattern occurred in an AJPS called Qinglongqiao, in which 10 more residential burglaries occurred during the pandemic days. In contrast, non-motor vehicle theft experienced a similar variation process. The offense decreased in 121 AJPSs, remained unchanged in 43 AJPSs, and increased in 19 AJPSs (Figure 4).

### 5.2. Regression Analysis

#### 5.2.1. NBR Analysis

The descriptive analysis showed that the occurrence of residential burglary and non-motor vehicle theft were suppressed and that their spatial concentrations were reduced during the pandemic period. The phenomenon indicated that crime opportunities were enormously controlled by the pandemic prevention-related measures. Hence, in order to further uncover how pandemic-related measures impact the occurrence of crime, a regression analysis was carried out to examine whether the associations between crime and facilities varied across the pre- to pandemic periods. First, both types of crime being aggregated by the AJPSs were examined using the K–S (Kolmogorov–Smirnov) test. The findings showed that *p*-values were all less than 0.05, which proved that the crimes distributed by the AJPS did not obey normal distribution, while the variance of the sample was larger than the mean value, so using the NBR model to analyze the data was supported. Following that, all independent variables (POIs, road network density, subway station flow) aggregated by the AJPS were simultaneously tested by using the VIF (variance inflation factor) method to see whether multicollinearity existed among the variables. The results indicate that all independent variables were less significantly correlated, so they could be input into the NBR model directly. In order to better detect the attractiveness of the urban facilities to crimes, green parks, open-air squares, train and shuttle bus stations, subway stations, and bus stops were removed from the independent variable list of regression for residential burglary as they are not the typical and appropriate targets for residential burglary offenders.

The regression was run using STATA and the results are shown in Table 5. It could be seen that the impact of urban facilities on both crimes demonstrated an obvious difference. First, seven significant independent variables (*p* < 0.05) were identified for residential burglary that occurred during the pre-pandemic period, but only specialized hospitals, gym rooms, arts and museums, and road density for vehicles demonstrated positive associations with crime, while low-priced hotels, temples and churches, and road density for only pedestrians showed a negative impact on crime. However, during the pandemic period, specialized hospitals, low-priced hotels, arts and museums, temples and churches, and road density for vehicles were no longer significant, and only the gym rooms and road density for pedestrians maintained a stable impact on crime. The comparison of the regressions demonstrated that most facilities lost their impact on residential burglary during the pandemic period.

Comparatively, for non-motor vehicle theft, five variables demonstrated a significant impact on offense during the pre-pandemic period, and community clinics, kindergartens, and road density for vehicles were positively associated with the offense, while wholesale markets and road density for only pedestrians demonstrated a negative effect. While during the pandemic period, the impact of wholesale markets, kindergartens, and road density for only pedestrians on crime disappeared, community clinics and road density for vehicles maintained a stable impact on crime, and more importantly, middle schools, bars and pubs, KTVs, and bus stops became new ones that significantly decided whether the crime occurred during the pandemic period.

The findings from the NBR model proved that although the significant impact of facilities on crime was observed during the pre-pandemic period, some of their impact on the offenses was reduced by the pandemic prevention-related measures during the pandemic period. For example, because entertainment activities were ordered to cease, gym rooms were empty and became potential positive attractors to residential burglary offenders, and as no customers presented, the bicycles and electric-driven bicycles concentrated around the bars and pubs, KTVs and community clinics were unguarded and crime opportunities appeared for offenders during the pandemic period. Moreover, as public traffic flow was reduced to a low level to prevent the spread of the virus, the passenger flow playing natural guardianship around bus stops was decreased and the bicycles that concentrated around the sites became suitable targets for offenders. Additionally, it was interesting to see that road density maintained a stable impact on both property crimes, but varied by the crime type. The road density for only pedestrians was negatively associated with residential burglary during both periods, which indicated that more accessible areas maintained a stable suppression effect on burglary regardless of the changes in the social environment. While the road density for vehicles was observed to be positively correlated with the occurrence of non-motor vehicle theft, this implies that bicycle-related offenses are more likely to occur within the areas with high vehicle flow concentrations.

#### 5.2.2. GWR Analysis

The NBR regression analysis proved that the impact of urban facilities on crime varied from the pre- to pandemic period, but simultaneously, the impact from some facilities on crime occurrence remained stable. In order to better examine whether the variables had a stable impact on crime in space, the dependent and independent variables were input into the GWR model and run using ArcGIS 10.7. The regression findings are shown in Table 6 and indicate that the GWR regression achieved a better fitting effect as the Adj R^2^s were more improved than the NBR model.

The coefficient distribution of the identified stable independent variables was visualized in maps to detect whether the impact of the facilities on crime varied in space. Figure 5 and Figure 6 demonstrate the regressed coefficient distribution of gym rooms and road density for pedestrians from GWR with residential burglary. The spatial distribution of the regression coefficients indicated that although the independent variables showed, on the whole, a significant impact on crime and maintained stability across the periods, their impact varied by the AJPS unit. First, it was observed that gym rooms were apparently positively associated with residential burglary across whole districts during the pre-pandemic period but more strong positive associations existed within the AJPSs located at the west, south, and east boundaries. However, during the pandemic period, the strong correlations were only kept within the AJPSs located on the northwest side of the urban main district of Beijing (Figure 5). The road density for pedestrians was proved to be negatively associated with residential burglary in the NBR model, but its impact on crime was not uniformly distributed across space, and it could be seen in Figure 6 that during both periods, the negative associations were mainly distributed within the center areas while some positive associations existed within the AJPSs on the peripheral areas. However, the distributions varied from the pre- to pandemic periods as positive impacts appeared within more AJPSs located on the west side of the urban district during the pandemic period (see Figure 6).

Figure 7 and Figure 8 demonstrate the regressed coefficient distribution of community clinics and road density for vehicles from GWR with non-motor vehicle theft. In Figure 7, it can be seen that the community clinics maintained a relatively stable spatial attractiveness to non-motor vehicle theft in both periods at the local spatial level, but the distribution of the regression coefficients changed within some peripheral AJPSs. Additionally, the road density for vehicles that was proven to be positively associated with crimes in the NBR model also impacted crime heterogeneously in space. It could be observed that during both periods, the strong positive impacts concentrated within the AJPSs located on the northwest side of urban districts, while some negative associations were also observed to exist within a few AJPSs located on the east side. However, during the pandemic period, more AJPSs located in the west area appeared to have a strong positive impact.

## 6. Conclusions

Aiming to uncover how crimes associated with different urban facilities varied by the pandemic-related prevention measures, this paper executed a practical study on residential burglary and non-motor vehicle theft in Beijing. By treating the main urban districts as the targeted area and the AJPS as the spatial unit, the crime that occurred during the FERPH period in 2020 and corresponding temporal period in 2019 as well as 29 types of urban facilities, which were assumed to have potential attractiveness or impact on property crimes, were collected. Following this, a rough descriptive analysis and regression analysis were conducted and several major findings were obtained.

(1)The results indicate that both types of property crime were significantly reduced during the pandemic period, which infers that the occurrence of the pandemic and its related prevention measures had an important negative impact on crime in Beijing. Furthermore, the concentrations of crimes in space were reduced and hot areas observed during the pre-pandemic period disappeared during the pandemic period.(2)The variations in the impact of urban facilities on residential burglary and non-motor vehicle theft across the pre- to pandemic periods were observed. Specifically, a couple of facilities that previously posed a significant impact on the occurrence of residential burglary lost their impact during the pandemic period, and some maintained stability at a significant level. While the phenomenon also occurred for non-motor vehicle theft, it was observed that some facilities (bars and pubs, KTVs, and bus stops) became significant ones conducive to the occurrence of crime during the pandemic period.(3)The stable variables that maintained a significant impact on crime during pre- and pandemic periods were also investigated at the AJPS level. The findings indicate that the impact of the variables on crime was heterogeneous in space and kept some variations across the pre- to pandemic period.

This study, although based on an investigation in Beijing, still has some implications for environmental criminology in theoretical and practical aspects. First, at the theoretical level, this work proves that the environmental variation of some urban facilities significantly changed crime occurrence, which supports the opinions of crime opportunity theories and allowed for a reduction in crime during the pandemic period to become interpretable. Next, in the practical aspect, the study identified vulnerable facilities that are more exposed to the risk of crime, thus consequently, the crime prevention measures could be deployed more scientifically in the backdrop of pandemic prevention.

Although this paper compared the impact of urban facilities on property crime between the pre- and pandemic periods, there are still some questions that need answers. For example, some facilities that had been assumed to be attractors for property crime actually showed a negative impact. According to the principle of routine activity opinion [4], whether the crime would occur was determined by the intersection of potential targets, motivated offenders, and the presence of guardianship, while the offenders play the vital role that directly determines whether the crime is generated, so how the facilities impacted on crime occurrence in conditions of strict social control should be further investigated combined with offender behavior, for example, journey-to-crime.

## Figures and Tables

**Figure 1 ijerph-20-02163-f001:**
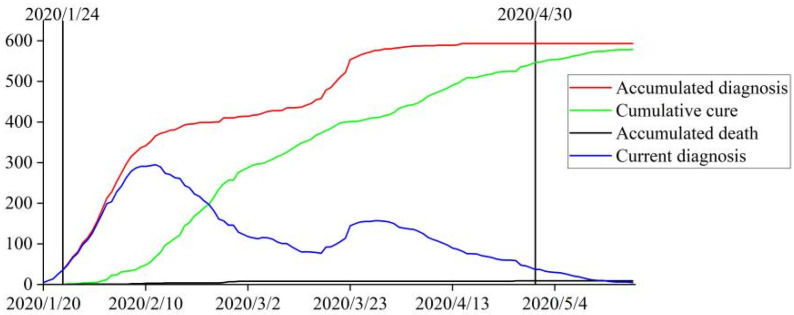
Temporal development of COVID-19 virus infection in Beijing.

**Figure 2 ijerph-20-02163-f002:**
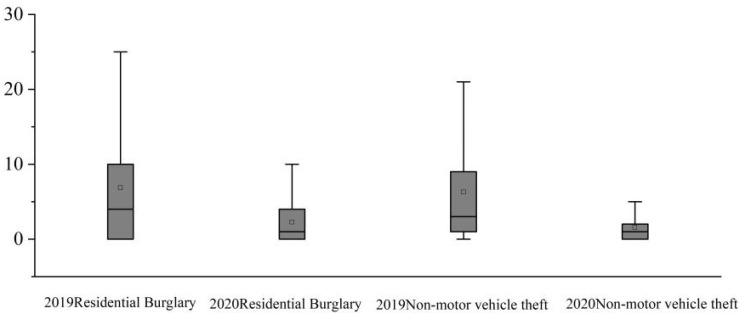
Boxplots of the residential burglary and non-motor vehicle theft count by day during the pre- (2019) and pandemic (2020) periods.

**Figure 3 ijerph-20-02163-f003:**
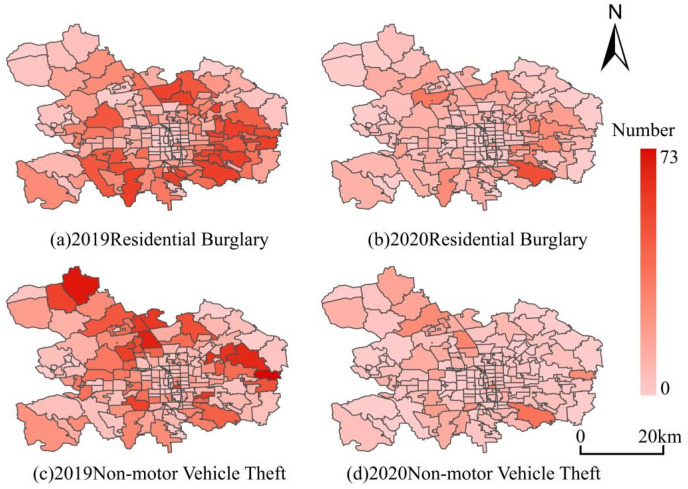
Spatial distribution of residential burglary and non-motor vehicle theft measured by AJPSs during the pre- (2019) and pandemic (2020) periods.

**Figure 4 ijerph-20-02163-f004:**
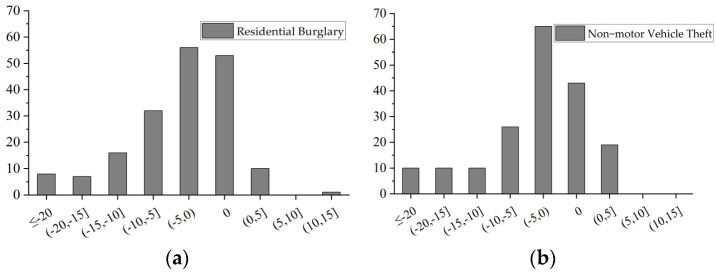
Quantitative variations on residential burglary and non-motor vehicle measured by AJPSs during the pre- and pandemic periods. (**a**) Residential burglary, (**b**) Non-motor vehicle theft.

**Figure 5 ijerph-20-02163-f005:**
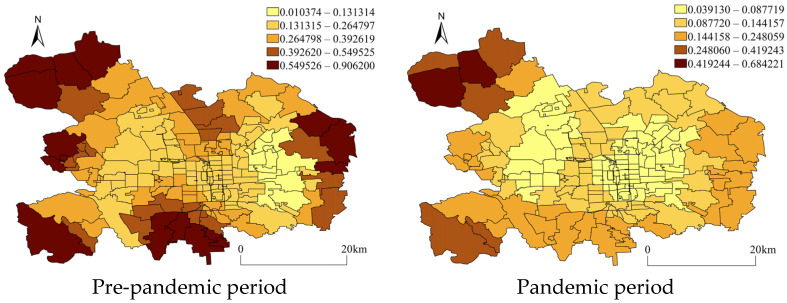
Coefficients of gym rooms from GWR with residential burglary. The coefficients were categorized by using natural breaks.

**Figure 6 ijerph-20-02163-f006:**
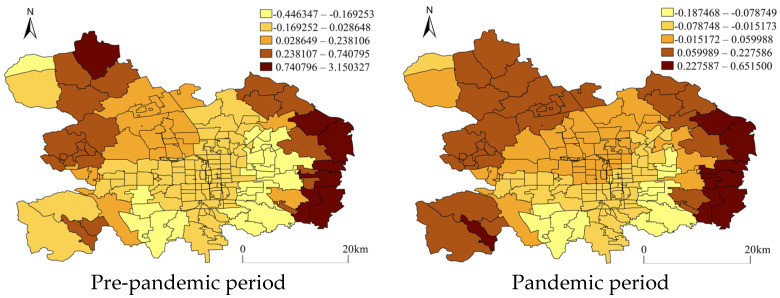
Coefficients of road density for pedestrians from GWR with residential burglary. The coefficients were categorized by using natural breaks.

**Figure 7 ijerph-20-02163-f007:**
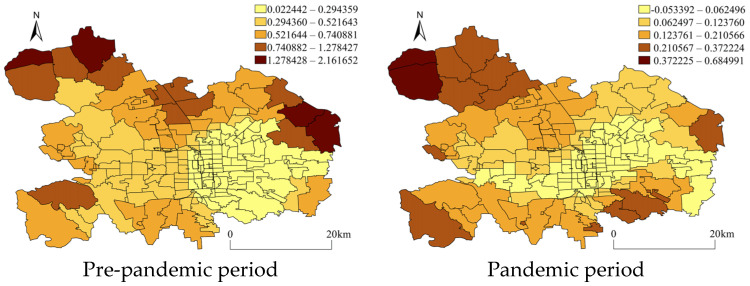
Coefficients of community clinics from GWR with non-motor vehicle theft. The coefficients were categorized by using natural breaks.

**Figure 8 ijerph-20-02163-f008:**
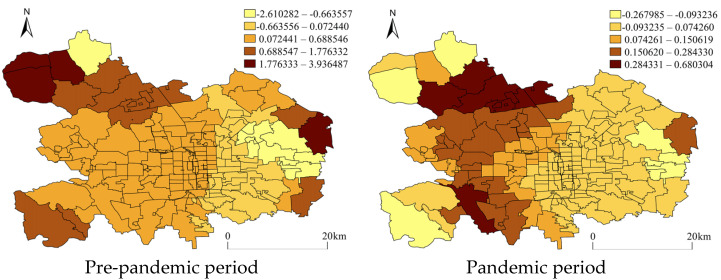
Coefficients of road density for vehicles from GWR with non-motor vehicle theft. The coefficients were categorized by using natural breaks.

**Table 1 ijerph-20-02163-t001:** The COVID-19 prevention and control measures implemented in urban facilities in Beijing.

Urban Facilities	Measures
Public transportation	Inter-city transport was stopped, maximum subway flow inner-city was limited no more than 50%.
Municipal industries	Only the industries necessary for urban operation (water, power, oil and gas supply, communications, municipal administration, etc.) were maintained.
Finances & Enterprises	Work switched from offline to online, working in the office was not encouraged.
Retails & Supermarkets	Supermarkets, food production and supply, logistics and distribution etc. worked as usual.
Hotels & Motels	Disinfection, temperature monitoring, and real-name inspection every day. Swimming pools and meeting rooms were closed, and hall dining was changed into room delivery.
Medical services	The patients with fevers and emergencies were of priority. Appointments for registration and online consultations were encouraged. Industries necessary for COVID-19 prevention and control (drugs, protective equipment, medical device production, transportation, and sales) were maintained.
Education agencies	All universities, high schools, middle schools, primary schools, and kindergartens as well as training institutions were closed.
Residential areas	Crowd gatherings were prohibited, unnecessary entrances and exits of communities were closed, and all opened entrances and exits were guarded to register visitors. Couriers and takeaways were not allowed to enter the residential area.
Entertainment places	All cinemas, Internet cafes, indoor and outdoor sports, and fitness venues were closed.
Restaurants	All indoor dining places were closed. Staff canteens’ dining hours were extended to prevent congestion.
Green lands & parks	All green lands and parks were open to the public.

**Table 2 ijerph-20-02163-t002:** Statistics of residential burglary and non-motor vehicle theft during the pre- and pandemic periods.

Crime Type	Temporal Variable	Mean	Std	Min	Max
Residential burglary	Pre-pandemic period	6.858	8.699	0	47
	Pandemic period	2.273	3.182	0	20
Non-motor vehicle theft	Pre-pandemic period	6.284	9.072	0	73
	Pandemic period	1.546	2.448	0	14

**Table 3 ijerph-20-02163-t003:** Statistics of urban facility by AJPS unit.

Independent Variable	Mean	Std	Min	Max
Shopping malls	2.88	3.25	0	16
Wholesale markets	22.61	26.45	0	188
Business office buildings	58.52	71.80	0	590
Banks & financials	16.24	16.77	0	102
Star rated hotels	4.42	4.97	0	28
Low-price motels	26.96	26.15	0	205
Villa areas	0.63	1.53	0	12
Factories & mills	2.90	4.20	0	23
Warehouses	18.31	21.26	0	173
Gasoline stations	1.91	2.42	0	12
Specialized hospitals	9.26	7.69	0	38
Community clinics	9.20	8.36	0	43
Welfare institutions	1.95	2.12	0	9
Universities & colleges	1.96	2.40	0	13
Middle schools	2.73	2.41	0	10
Primary schools	3.93	3.26	0	17
Kindergartens	8.52	8.41	0	47
Gym rooms	12.13	14.68	0	84
Arts & museums	10.19	11.03	0	85
Temples & churches	1.34	1.96	0	10
Bars & pubs	3.87	10.96	0	130
KTVs & nightclubs	3.22	3.60	0	18
Game halls	4.28	4.56	0	20
Amusement parks	2.11	3.53	0	26
Green parks	2.64	2.94	0	16
Open-air squares	1.45	1.78	0	8
Train & shuttle bus stations	0.37	1.11	0	12
Bus stops	14.31	11.48	0	50
Subway stations	0.68	0.88	0	6

**Table 4 ijerph-20-02163-t004:** Global Moran’s I statistics of residential burglary and non-motor vehicle theft during the pre- and pandemic periods.

AJPS, *n* = 183	Residential Burglary		Non-Motor Vehicle Theft	
	Pre-Pandemic	Pandemic	Pre-Pandemic	Pandemic
Global Moran’s I	0.291 ***	0.192 ***	0.121 ***	0.067 ***

*** *p* < 0.001.

**Table 5 ijerph-20-02163-t005:** The NBR regression results of residential burglary and non-motor vehicle theft during the pre- (2019) and pandemic (2020) periods.

**Independent Variable** **(AJPS, *n* = 183)**		Residential Burglary		Non-Motor Vehicle Theft	
		**2019**	2020	2019	2020
Control variables Urban facility variables	Intercept Subway flow Road density (vehicle) Road density (bicycle) Road density (pedestrian) Shopping malls Wholesale markets Business office buildings Banks & financials Star rated hotels Low-price motels Villa areas Factories & mills Warehouses Gasoline stations Specialized hospitals Community clinics Welfare institutions Universities & colleges Middle schools Primary schools Kindergartens Gym rooms Arts & museums Temples & churches Bars & Pubs KTVs & nightclubs Game halls Amusement parks Green parks Open-air squares Train & Shuttle bus stations Bus stops Subway stations AIC Adj *R^2^*	0.39 −0.02 0.07 ** 0.01 −0.04 * −0.03 0.00 0.00 −0.01 −0.01 −0.01 ** −0.04 0.01 0.01 0.06 0.03 * −0.02 0.06 −0.03 0.02 0.02 0.03 0.03 ** 0.01 * −0.14 ** 0.00 0.05 0.04 −0.02 - - - - - 945.572 0.1730	−0.29 −0.64 0.03 0.02 −0.04 * −0.04 0.00 0.00 −0.01 0.01 0.00 −0.13 0.00 0.00 0.08 0.02 0.03 0.01 −0.03 0.06 −0.03 0.03 0.03* 0.01 −0.01 0.00 0.06 0.02 −0.01 - - - - - 682.208 0.1389	0.31 −0.06 0.07 * 0.04 −0.05 * −0.05 −0.01 * 0.00 0.00 0.05 −0.01 −0.12 −0.03 0.01 0.05 0.00 0.04* 0.06 0.03 0.03 −0.01 0.06 ** 0.00 0.00 −0.05 0.01 0.02 0.01 −0.05 0.07 −0.03 0.06 0.02 0.10 1000.01 0.1091	−1.03 −0.36 0.07 * 0.04 −0.03 −0.08 0.01 0.00 −0.01 0.03 0.00 −0.04 −0.02 0.00 −0.08 −0.01 0.04* 0.02 0.03 −0.14 * 0.00 0.02 −0.02 −0.01 −0.06 0.01 * 0.09 * −0.06 0.01 0.07 −0.09 0.02 0.04 * 0.32 610.381 0.1205

* *p* < 0.05; ** *p* < 0.01.

**Table 6 ijerph-20-02163-t006:** AICs and Adj R^2^s derived from GWR regression to residential burglary and non-motor vehicle theft during the pre- and pandemic periods.

AJPS, *n* = 183	Residential Burglary		Non-Motor Vehicle Theft	
	Pre-Pandemic	Pandemic	Pre-Pandemic	Pandemic
AIC Adj R^2^	1197.25 0.53	891.75 0.34	1254.27 0.42	800.59 0.33

## Data Availability

Due to an agreement with the crime data provider, we are not allowed to disclose the crime data.

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
