# Peer review of "The Impact of Urban Facilities on Crime during the Pre- and Pandemic Periods: A Practical Study in Beijing"

_ijerph, 2023, doi:10.3390/ijerph20032163_

Round 1
Reviewer 1 Report
Overall, the manuscript is a solid and well-organized paper, which attempts to investigate how the urban facility impact the residential burglary and non-motor theft during the pre and post COVID times, and how the impact change over the time. Several comments / questions are listed below.
1. In line 255, the author states “the pandemic posed a significant negative effect on property crime's occurrence.”, it would be better to have some numbers to show the statistically significant.
2. The paper is well written, but it seems the contribution of this paper is not so clear, it would be better to add 1 or 2 paragraphs to summary the contribution of this paper, both theoretically and practically.
Minor suggestions
1. In Line 203, for the “Amap”, it would be better to add a little bit details, not everyone know what is Amap.
Author Response
Overall, the manuscript is a solid and well-organized paper, which attempts to investigate how the urban facility impact the residential burglary and non-motor theft during the pre and post COVID times, and how the impact change over the time. Several comments / questions are listed below.
Response: thanks again for the comments.
1.In line 255, the author states “the pandemic posed a significant negative effect on property crime's occurrence.”, it would be better to have some numbers to show the statistically significant.
Response: We run the independent sample t-test to examine the difference of crime count by day during the pre- and pandemic periods, and the results were all significant at p<0.001 levels. It has been added into the text, see line 254 to 257 on page 7.
2.The paper is well written, but it seems the contribution of this paper is not so clear, it would be better to add 1 or 2 paragraphs to summary the contribution of this paper, both theoretically and practically.
Response: Suggestions accepted. We added a paragraph indicating the theoretical and practical contributions of this paper in Conclusion sections. See line 433 to 440 on page 14.
Minor suggestions
1.In Line 203, for the “Amap”, it would be better to add a little bit details, not everyone know what is Amap.
Response: We added a footnote on page 6 to indicate the basic introduction of AMap, see footnote on page 6.
Reviewer 2 Report
You note in your conclusion that one shortcoming of your research is that you did not factor in the matter of ambient population. I am not sure that it is a shortcoming. Specifically, you found that most of the facilities ceased to be hotspots post-pandemic. That tells me that relative to the crime decline that was occurring in Beijing generally post-pandemic, the facilities experienced a greater decline. In other words you have demonstrated the impact selected facilities have on crime. That is no doubt a consequence of the human traffic to and at those facilities, but that is another matter....and an obvious one. Simply, as you know, increased and decreases in population in any area will generally cause changes in the crime rate (all else being equal). On another note you use the term "post-pandemic" but in reality you mean "during the pandemic period". I know what you mean, but to be more accurate, I would suggest a term other than post-pandemic. For example you might say "before and during pandemic period". Still, I see this as a good article which can be helpful to community planning and police resource allocation. Nice job.
Author Response
You note in your conclusion that one shortcoming of your research is that you did not factor in the matter of ambient population. I am not sure that it is a shortcoming. Specifically, you found that most of the facilities ceased to be hotspots post-pandemic. That tells me that relative to the crime decline that was occurring in Beijing generally post-pandemic, the facilities experienced a greater decline. In other words you have demonstrated the impact selected facilities have on crime. That is no doubt a consequence of the human traffic to and at those facilities, but that is another matter....and an obvious one. Simply, as you know, increased and decreases in population in any area will generally cause changes in the crime rate (all else being equal).
Response: Thanks for the comments, we removed the descriptions of the shortcomings from the last paragraph in Conclusion sections, instead, we believed the following work should focused on offender’s target searching behavior because some facilities which had been assumed to positively attract crime showed negative effect, thus in future work the study to offenders’ behaviors in pandemic background should be included. The indications were inserted into the text, see line 442 to 449 on page 14.
On another note you use the term "post-pandemic" but in reality you mean "during the pandemic period". I know what you mean, but to be more accurate, I would suggest a term other than post-pandemic. For example you might say "before and during pandemic period". Still, I see this as a good article which can be helpful to community planning and police resource allocation. Nice job.
Response: The “post-pandemic” was replaced by “pandemic” per reviewer’s suggestion.